# Diagnostic and Management Issues in Patients with Late-Onset Ornithine Transcarbamylase Deficiency

**DOI:** 10.3390/children10081368

**Published:** 2023-08-09

**Authors:** Majitha Seyed Ibrahim, Jessica I. Gold, Alison Woodall, Berna Seker Yilmaz, Paul Gissen, Karolina M. Stepien

**Affiliations:** 1Department of Chemical Pathology, Teaching Hospital Batticaloa, Batticaloa 30000, Sri Lanka; 2Division of Human Genetics, Department of Pediatrics, The Children’s Hospital of Philadelphia, Philadelphia, PA 19104, USA; 3Adult Inherited Metabolic Diseases, Salford Royal Hospital, Northern Care Alliance NHS Foundation Trust, Salford M6 8HD, UK; 4Great Ormond Street Institute of Child Health, University College London, London WC1E 6BT, UK; 5Department of Paediatric Metabolic Medicine, Great Ormond Street Hospital for Children NHS Trust, London WC1N 3JH, UK; 6National Institute of Health Research, Great Ormond Street Biomedical Research Centre, London WC1N 1EH, UK; 7Division of Cardiovascular Sciences, University of Manchester, Manchester M13 9PL, UK

**Keywords:** late-onset OTC deficiency, urea cycle disorders, clinical manifestations, adults

## Abstract

Ornithine transcarbamylase deficiency (OTCD) is the most common inherited disorder of the urea cycle and, in general, is transmitted as an X-linked recessive trait. Defects in the OTC gene cause an impairment in ureagenesis, resulting in hyperammonemia, which is a direct cause of brain damage and death. Patients with late-onset OTCD can develop symptoms from infancy to later childhood, adolescence or adulthood. Clinical manifestations of adults with OTCD vary in acuity. Clinical symptoms can be aggravated by metabolic stressors or the presence of a catabolic state, or due to increased demands upon the urea. A prompt diagnosis and relevant biochemical and genetic investigations allow the rapid introduction of the right treatment and prevent long-term complications and mortality. This narrative review outlines challenges in diagnosing and managing patients with late-onset OTCD.

## 1. Introduction

The urea cycle is the physiological primary pathway for removing nitrogen, a toxic byproduct of amino acid metabolism. Consisting of five enzymes, one cofactor producer and two mitochondrial transport molecules in the mammalian liver, the urea cycle converts ammonia to urea for urinary excretion [1,2]. Urea cycle disorders (UCD) are monogenic disorders caused by decreased function in any of the eight components of this cycle. Common complications of UCD include the accumulation of ammonia, which is neurotoxic [3,4,5,6,7], and hepatic dysfunction. The most prevalent UCD is an ornithine transcarbamylase deficiency (OTCD) [8,9,10,11]. The anabolic OTC enzyme is responsible for the transfer of a carbamoyl group from carbamoyl phosphate to the amino group of L-ornithine, producing citrulline in an early step of the urea cycle [12,13]. 

OTCD was first described in 1962 by Russell regarding two girls at the age of 1 year and 8 months and 6 years [3]. The estimated prevalence is between 1:14,000 and 80,000 [14]. The X-linked inheritance of OTCD is unique with the remaining UCDs inherited in an autosomal recessive manner. Due to its X-linked recessive inheritance, OTCD tends to present earlier, and more severely, in males [11]. However, patient phenotype can vary due to random X inactivation in females and hypomorphic variants in OTC that cause a partial enzyme deficiency and later onset of symptoms in males [7,15]. While the other UCDs are included on newborn screening panels, the diagnosis of OTCD is challenging. The early presentation of the classical OTCD makes newborn screening unhelpful in these patients. In addition, the late-onset patients may be very difficult to diagnose on a metabolite analysis alone. There are no common mutations as such and therefore full gene sequencing needs to be completed and include a promoter analysis. 

Thus, we use age at the presentation and severity of the phenotype to classify patients into two categories, neonatal and late-onset. Current evidence estimates 30% of cases are neonatal onset and 70% are late-onset [16]. The neonatal presentation is severe [16], leading to coma and death [17]. It commonly presents during the first few days of life [18] and affects mainly hemizygous males with a complete enzyme deficiency [7,19]. This category is rare in females (7%) [20]. Neonates present with severe rapid-onset neurological manifestations, leading to death within the first few hours if untreated and within a few months despite medical management. Of the survivors, many develop severe neurological consequences [18].

Those with late-onset OTCD can develop symptoms anytime from infancy to later childhood, adolescence or adulthood [21]. Hemizygous males with a partial enzyme deficiency and around 20% of heterozygous females fall into this category. More than 80% of heterozygous females remain asymptomatic throughout life [4,8,9,22]. Clinical manifestations in this form of OTC are nonspecific and are mainly due to hyperammonemia. They include neurological (encephalopathy and migraine headaches), psychiatric (bipolar-disorder-like symptoms) or gastrointestinal symptoms (hepatic dysfunction and cyclic vomiting). All symptoms can present as a combination [23,24,25,26]. Symptoms can be triggered by stressors such as prolonged fasting, certain medications and pregnancy, and may lead to death if not treated immediately [27]. 

In this narrative review, we outline challenges in diagnosing and managing patients with late-onset OTCD. Prompt treatment of OTCD can be lifesaving but can only follow a successful diagnosis. An overview of the upcoming therapeutic developments for this rare condition is a novel aspect of the review. 

## 2. Literature Review

A systematic literature search was performed in Pubmed, Embase and CINAHL ending on 1 April 2023. Original studies, reviews, case reports, case series, conference abstracts and proceedings including patients with known late-onset OTCD were eligible for inclusion. Data from abstracts that were not written in English were excluded. 

The data extraction was performed from each full text by two authors (M.I. and K.S.).

The search was performed using combinations of several keywords: “Ornithine Transcarbamylase deficiency”, “pregnancy”, “transition”, “therapy”, “gene therapy”, “dietetic modifications”, “clinical outcome”, “mortality”, “acute decompensation”, “laboratory investigations” and “genetic counselling” with Boolean operators “AND” and “OR” for a thorough search. We performed a descriptive narrative synthesis. 

## 3. Diagnosis

### 3.1. Clinical Manifestations as Diagnostic Clues

Recognizing the symptoms of late-onset OTCD is challenging because it is a rare disease with a highly heterogeneous, episodic presentation [19,24,28]. Therefore, the prompt diagnosis and the rapid introduction of dietetic modifications, followed by appropriate treatment with ammonia scavengers, are essential to reduce the mortality and improve the quality of life [17]. 

Clinical manifestations of adults with OTCD vary in acuity. Gastrointestinal symptoms include vomiting, abdominal pain and hepatic dysfunction. Patients may present with chronic migraine headaches or neurological signs of hyperammonemia such as lethargy, disorientation, agitation, confusion, reduced consciousness and prolonged generalized seizures. An initial presentation with psychiatric symptoms can occur and OTCD should be considered in adolescents with new-onset psychosis or bipolar-like symptoms [21,26,27,29,30,31,32]. 

### 3.2. Triggering Factors

Clinical symptoms can be aggravated by metabolic stressors (an increased protein consumption including the use of sport supplements, infection, trauma, fever, severe illness and acute stress) or presence of a catabolic state (intrapartum and immediate postpartum, menstruation, gastric bypass surgery, starvation and fasting), or due to increased demands upon the urea cycle (systemic corticosteroids, rapid weight loss, gastrointestinal bleeding trauma and chemotherapy) [19,21,26,27,31,33,34,35,36]. Apart from steroids (dexamethasone, hydrocortisone and methylprednisolone) and chemotherapy, anticonvulsants (valproate and topiramate), metoclopramide and azathioprine may increase demands on the urea cycle, thus triggering an initial OTCD presentation [37]. 

In addition, underlying morbidities such as cardiac dysfunction and nonalcoholic fatty liver disease also unmask the OTCD in adults [24]. As hyperammonemia may become apparent in these patients during acute metabolic decompensation [21], it is important to consider and initiate the relevant investigations. 

### 3.3. Laboratory Work Up

As the clinical presentation may be non-specific, the diagnosis should be confirmed with the biochemical and genetic testing [4]. A patient presenting with the symptoms listed above should have a blood ammonia level measured, using a sample collected through a free-flowing venipuncture and placed immediately on ice. Ammonia levels should also be drawn from any patient presenting with encephalopathy and respiratory alkalosis as hyperammonemia irritates the central respiratory drive and increases the respiratory rate. Late-onset OTCD is not uncommon to appear in adults with hyperammonemia, especially with no background of underlying liver cirrhosis [21,38]. 

Once hyperammonemia is established, diagnostic biochemical investigations including plasma amino acids and urine organic acids should be performed. Plasma amino acids will aid the diagnosis—showing high plasma glutamine and low plasma citrulline—and urine organic acids should show orotic aciduria [13,24,25,33,39]. The urine extraction of orotic acid results from excess carbamoylphosphate, which diverts into the pyrimidine pathway [16]. Orotic acid quantification in urine following allopurinol or protein loading, and/or a pedigree analysis, can be used as an alternative for female carriers [27]. However, the sensitivity and specificity of the allopurinol test is 91% and 70%, respectively. As protein loading can trigger symptoms, it is better to avoid it [29]. Orotic acid concentration can, however, be raised in other conditions such as lysinuric protein intolerance, Rett syndrome, certain forms of liver disease, certain forms of cancer and secondary to the use of certain medications and alcohol [40].

Laboratory investigations may only be diagnostic during an acute, symptomatic phase, therefore increasing the importance of a genetic diagnosis [12,27]. Approximately 80–90% of cases are detected through a mutation analysis [25,27,41,42]. In some laboratories, the likelihood of not confirming the mutation in a biochemically confirmed OTC family is as high as 30% (personal observation). Functional assays including enzyme activity in the liver and intestinal mucosa are helpful [27], although they are more invasive and require a specialized biochemical laboratory. In most males with the absence of mutation, an analysis of liver enzymes is diagnostic. However, functional studies may be unreliable due to random X inactivation leading to a mosaic distribution of residual OTC activity [43]. 

With advances in gene sequencing technology, novel mutations are detected over time [15]. More than 500 disease causing mutations have been reported in the OTC gene so far [44]. Of all, single base substitutions account for 70–84%, 12% are small-fragment deletions or insertions and the rest are large fragment deletions [14,44]. Several mutations have been reported in the promoter and enhancer region [41]. New high-throughput assays are in development to aid in classifying variants by functional status [45]. In general, the complete dysfunction of enzymes is due to the distortion of the reading frame of the coding sequence via nonsense, insertion and deletion. The substitution of an amino acid in either the active site or the hydrophobic core leads to the partial dysfunction of the enzyme [14,46]. Approximately 60% of hemizygous males with the severe neonatal form have mutation around the active site of the enzyme. The rest show mutation at the periphery, present later with less severe manifestations [8]. However, this classification is not definitive as presentation can vary among two individuals with the same OTC variant [46,47]. 

## 4. Management

### 4.1. Common Complications in Adults

The management of patients with OTCD focuses on the prevention of both acute and chronic complications [36]. Acute hyperammonemia leads to encephalopathy and neuronal death. Glutamatergic receptors in the brain become overstimulated by excess glutamate and glutamine, which affects neuronal function. During chronic hyperammonemia, increased GABAergic signaling results in reduced neurocognitive functions [36]. 

Consequently, OTCD patients frequently show long-term neuropsychological complication such as a learning disability, intellectual disability, attention deficit disorder and executive function deficit [29,48,49] and chronic psychiatric problems [19]. Fine motor, executive and cognitive flexibility and inhibition ability functions were shown to be affected in asymptomatic and symptomatic patients but no measurable differences were noted in attention, language or verbal memory [48,49]. Late-onset OTCD patients may present with lethargy, “fogginess of their brain” or psychosis. As a result of persistently raised plasma ammonia, the cognitive function may be affected, which becomes more apparent during acute illness when plasma ammonia raises significantly higher than usual. In such situations, their capacity to make decisions is limited. Any clinical decisions regarding the patient’s health, e.g., procedures and scans under sedation, are made by the medical team in the patient’s best interest. 

### 4.2. Multidisciplinary Team

The involvement of the multidisciplinary team (MDT) is very important for the successful management [49,50]. This MDT comprises an intensive care team, a neurologist, a surgical team, a nutritionist, a metabolic specialist, pharmacy staff, social workers, a learning disability nurse, a hepatologist and genetic counselors [2]. The multispecialty and multidisciplinary approach facilitates an early diagnosis and comprehensive management, and results in positive clinical outcomes. 

### 4.3. Dietetic Support

In a significant number of cases, patients are protein averse and often follow a vegetarian diet or only consume small quantities of high-biological-value proteins such as meat and fish (a natural protein intake not higher than 0.8 g/kg of body weight). A detailed diet history is important to understand current nutritional intake and to inform dietetic advice individually tailored to the needs of the patient.

If patients are not protein averse, then they are advised to limit the intake of protein. Recommendations on protein intake are informed by current protein intake, blood tests including ammonia and plasma amino acid concentrations and clinical symptoms. If protein intake is too low, a total protein or an essential amino acid supplement may be recommended if dietary intake cannot be adjusted to meet the patient’s needs. A reduction in protein intake will be advised if ammonia is raised above the recommended range or if high ammonia is indicated with clinical symptoms in the absence of an ammonia result.

Due to their restricted diet, patients can be at risk of deficiencies in iron, zinc, copper, calcium, cobalamin and essential fatty acids. Key micronutrients should therefore be monitored, and replaced if below the normal range. It is of particular importance to monitor zinc, which acts as a cofactor for the OTC enzyme [50].

Tube feeding (gastrostomy) is more common in individuals with early-onset OTCD, to ensure the adequate intake of energy and protein. Indications for a nasogastric tube in acute OTCD include difficult swallowing, the refusal of food, gastrointestinal discomfort, a poor palatability of drugs and supplements if required. A gastrostomy is recommended when tube feeding is to be continued for more than a short time and/or overnight feeds are required [50]. 

### 4.4. Family Screening—Genetic Counsellors

The genetic counselor role is pivotal in educating the family in the inheritance of this rare disease and to help them to identify other members at risk with genetic testing [2]. Genetic testing of newborn offspring with OTC carriers is useful to genetic counselling and to inform early intervention [51]. The counselling needs to include experts in OTCD because it is impossible to predict the severity of disease in female carriers. The clinical picture in heterozygous females varies among individuals both in terms of the onset of the disease and its severity, which results from the genotype and the degree of inactivation of the mutated X chromosome in hepatocytes. Up to 85% of heterozygous females do not develop symptoms of hyperammonemia during their lifetime [37]. In rare cases, de novo mutations have been reported in up to 26% of male probands [52], meaning their mothers are not carriers for the mutation. 

### 4.5. Transition from Pediatric to Adult Hospital

With the advances in diagnostic technology and therapeutic options, affected children reach adulthood. Therefore, a well-driven, planned transition of adolescents and young adults from child-centered care to an adult-oriented care system is essential. Successful transition will lead to continuity in care and provide metabolic stability [53]. In the process of transition and the transfer of care, it is essential to identify whether the young adult recognizes potential triggers of a hyperammonemic episode and would know how to seek medical care for concerning symptoms. 

However, different challenges are faced in the management and social requirements vary from childhood to adulthood [54,55]. A lack of guidelines on the transition care leads to anxiety among young adults and their families [56,57]. 

The required facilities and adequate specialist training of adult care physicians in Inherited Metabolic Diseases (IMD) management are difficult to access in most countries and non-existent in some [55]. As a result, young adults remain under pediatricians’ care long-term or, in some cases, are lost to follow up [57,58].

The common challenges around the transition to adult services include autonomy and becoming independent, which can lead to a poor adherence to the medicine and diet [56]. An additional challenge in transition is the possible use of alcohol and illicit substances. These can increase the risk of decompensation events. Changes in dose and frequency of medication and the volume of diet are influenced with age, growth spurts, the magnitude of the disease and associated co-morbidities. Limitations in the availability of amino acid supplements and a dietetic regimen can form obstacles to engagement during the transfer of care, as can changes of medications in the adult hospital [24,53]. 

A study by Ladha et al. [53] has shown that the total average transition readiness assessment questionnaire (TRAQ) score for subjects with UCDs, including OTCD, in this cohort was low at 2.96/5.0, which was significantly lower than the published TRAQ score from young people without any ongoing debilitating conditions [53]. It was concluded that the individuals with a UCD and low IQ may benefit from personalized education in keeping with domains of the TRAQ [53].

OTCD patients may have a wide range of cognitive impairment and behavioral problems. Further arrangements and refinements should be considered for patients who have a poor neurocognitive outcome in order to have a successful transition [55,57]. A tailored approach is necessary for timing, planning and implementing the transition for each patient.

For individuals with late-onset OTCD that have been mostly asymptomatic in their youth and adolescence, and who, in many cases, are not required to follow even an emergency regimen during illness, the underlying diagnosis may have a lower impact on their overall knowledge about transition. 

### 4.6. Preimplantation Genetic Diagnosis

A preimplantation genetic diagnosis (PGD) is more beneficial over a prenatal diagnosis either with chorionic villous sampling or genetic amniocentesis with molecular testing for unfortunate families. PGD allows for the preselection of an OTC-free embryo for implantation and prevents the unnecessary therapeutic termination of the pregnancy, mortality of offspring and intense trauma to the family [23,34,59,60,61]. 

### 4.7. Pregnancy 

Pregnancy is a risk for hemizygous females because the catabolic state, both intra- and post-partum, is a critical trigger for hyperammonemia [36,62,63]. Anorexia, nausea, vomiting and dehydration in the first trimester increase the risk of catabolism and it should be managed aggressively both medically and dietetically, in the form of extra calories, fluids and medications to manage nausea and vomiting. Management can be intravenous if indicated [23,63]. Protein levels rise during the immediate post-partum period due to uterine involution and patients must be managed carefully to prevent hyperammonemia. This management would usually include a protein restriction, to improve ammonia levels, and additional calories, to reduce the complicating risk of catabolism.

Uncommon clinical manifestations such as psychosis during post-partum can also appear in OTCD [29]. However, it should not be misdiagnosed as post-partum depression. The diagnosis of OTCD in mothers has been previously made after their sons developed symptoms of an acute illness and hyperammonemia [23,34,62]. 

A lack of proper planning and limited time for multidisciplinary work are main risks of the unexpected OTCD diagnosis during pregnancy [21,34]. It may result in detrimental consequences for the mother and their baby. 

### 4.8. Mortality

Mortality is a key indicator for the further evaluation of the diagnosis of the late-onset phenotype [21,28]. A meta-analysis by Burgard et al. (2016) has shown that all UCDs, apart from females with OTCD, have a high risk of early-onset manifestations and neonatal death [20]. Despite significant diagnostic and therapeutic improvements, the mortality rate has not changed in several decades. In older patients, the risk of mortality is very high if high ammonia is not identified and early treatment is not initiated [64]. The initial peak value of >1000 µmol/L shows a higher mortality compared to <500 µmol/L [21,65,66]. The family history is often very relevant, as the diagnosis of late-onset OTCD can be made in an adult person after a grandchild presents with a severe illness. As an example, the late presentation of OTCD in a 62-year-old man did not raise a suspicion of a rare disease until their grandson was confirmed to be affected with the same mutation. The delayed diagnosis was made post-mortem with an undetectable enzyme activity in their liver [64]. 

Reported causes of mortality in late-onset OTCD include hyperammonemic encephalopathy, cerebral edema and cerebellar herniation, an elevated intracranial pressure (ICP) and status epilepticus [20,22,23,32]. 

## 5. Current and Upcoming Therapies 

For acute management, three points are to be considered—the removal of ammonia, reversing the catabolic state and avoiding exogenous protein and the initiation of nitrogen scavengers [4,12,27,30,35,43,67,68]. 

The removal of ammonia with hemofiltration is the recommended choice before transferring to a specialized center when the ammonia level is >200 µg/dL [26,28,67]. The provision of caloric support with 10–20% of IV glucose, and SMOF or intralipids, if required to achieve the calorie requirement, is necessary to prevent the catabolism [67,68]. An exogenous protein supply should be avoided temporarily but not for more than 48 h. Restarting protein intake is then recommended to prevent the endogenous protein catabolism [67].

If the dietetic management is not sufficient, then sodium phenylacetate and sodium benzoate are used as nitrogen scavengers with L-citrulline (oral preparation) or L-arginine hydrochloride (IV and oral preparation) supplementation [4]. Glycerol phenylbutyrate has been shown to improve the overall metabolic control [69]. Ammonia is eliminated via an alternative pathway by sodium phenylbutyrate and sodium benzoate. Arginine activates the urea cycle [8,26] and also controls the proteolysis and in turn reduces the urea production [68]. 

Long-term management mainly focuses on a nutrient-controlled diet, especially with protein, the supplementation of L-arginine and L-carnitine, ammonia scavengers whenever indicated [68] and the avoidance of triggers [17]. L-carnitine crosses the blood–brain barrier and helps reduce the level of ammonia through reaction cascades [28]. 

Orthotopic liver transplantation (OLT) is an option for patients who are affected by severe, recurrent attacks or failure for the medications to reduce ammonia. Patients who underwent liver transplantation showed a good outcome and improved quality of life [70]. The long-term neurocognitive outcome needs to be closely monitored after liver transplantation to provide the best supportive care [71]. 

### 5.1. Hepatocyte Transplantation

This mode of therapy is a substitute for OLT but, whilst it shows less surgical complications, the effect is transient, and frequent repeat hepatocyte transfusions are required as well as similar levels of immunosuppression [72]. Mesenchymal stem cells that differentiate into hepatocytes theoretically have more capacity to proliferate in a diseased liver and are highly immunotolerant compared to mature implanted hepatocytes [52,73]. However, there is no evidence from clinical trials that this method works in humans [74]. 

### 5.2. Upcoming Therapies

OTCD is an appealing candidate for novel therapies including gene addition, mRNA therapy and genome editing [75]. The first gene therapy clinical trial targeting OTCD was initiated in the late 1990s using adenoviral vectors. This attempt ended with the death of one of the participants due to a severe immune response [76]. 

Efforts continued with safer viral vectors including adeno-associated virus (AAV) after the 2000s. Early pre-clinical proof of concept studies with hepatotropic AAV vectors showed metabolic correction and prolonged survival [77]. More recent studies showed that AAV-mediated gene addition can also prevent chronic liver damage and fibrosis [78]. A phase I/II safety and dose-finding study (CAPtivate, ClinicalTrials.gov Identifier NCT02991144) using AAV8 recruited adult patients and has been completed without any serious treatment-related adverse events. A long-term follow-up study (ClinicalTrials.gov Identifier NCT03636438) is ongoing. In total, 7/11 treated patients were determined as responders and remained metabolically stable from 2 to 4.5 years. Four complete responders have required neither ammonia scavengers nor dietary restrictions after treatment [79]. A phase III study is currently recruiting 50 participants who are 12 years and older (ClinicalTrials.gov Identifier NCT05345171).

As AAVs mostly do not integrate into the host genome and mainly remain episomal, AAV DNA is lost during the cell division. This is still an important hurdle to achieve sustained transduction in the growing liver [80]. To overcome this, engineered AAV capsids with a higher transduction efficiency have been developed. AAVLK03 was found to have 10 times higher transduction rates compared to AAV8 [81]. Therefore, it has been a strong candidate to treat OTCD and, after proving safety in preclinical studies, clinical translation is ongoing [82]. A phase I/II open-label, multicenter clinical trial (HORACE, ClinicalTrials.gov Identifier NCT05092685), aiming to treat pediatric patients, is at the pre-recruiting stage. 

mRNA technology is another promising gene therapy approach for OTCD. However, as it provides a rapid and transient expression of the protein, repeated administration is necessary for long-term efficacy [83]. The multi-dose systemic administration of mRNA provided metabolic correction as well as an improved survival in a murine model of OTCD [84]. A phase Ia randomized, double-blinded, placebo-controlled study (ClinicalTrials.gov Identifier NCT04416126) using a single dose of OTC mRNA has just been completed. A phase Ib study (ClinicalTrials.gov Identifier NCT04442347) is currently active and has recruited 12 adults. A phase II study (ClinicalTrials.gov Identifier NCT055260660 using multi-dose OTC mRNA is now recruiting both adolescents and adults.

Both in vivo and ex vivo genome editing strategies are emerging for OTCD. AAV-delivered CRISPR-Cas9-mediated gene editing was found to be effective both in a OTC-deficient murine model and primary human hepatocytes [85,86,87]. Ex vivo CRISPR-corrected human hepatocytes provided metabolic correction in a murine model [88]. This preclinical success is encouraging for clinical translation. 

## 6. Conclusions

The early diagnosis of late-onset OTCD is important as even asymptomatic patients are at risk for a life-threatening hyperammonemic crisis and can benefit from a tailored UCD management plan. Comorbidities and dietetic preferences may play a role in masking or unveiling a late-onset OTCD. The condition requires unique clinical care, including the prevention and treatment of acute metabolic decompensation; a patient-centered approach; continued education regarding symptom recognition and adherence to the therapy and diet; highly specialized management through pregnancy; and genetic counselling regarding the risks of the condition. There is a clinical need for diagnostic and disease monitoring biomarkers. Upcoming therapies will bring meaningful clinical benefits to the patients and improve their quality of life. An increase in residual enzymatic OTC activity above the therapeutic target of 5% might reduce mortality and the incidence of a severe OTCD phenotype. 

## Data Availability

Not applicable.

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
