# Peer review of "Diagnostic and Management Issues in Patients with Late-Onset Ornithine Transcarbamylase Deficiency"

_children, 2023, doi:10.3390/children10081368_

Round 1

Reviewer 1 Report

Overall this is a well written narrative review of late-onset OTC deficiency. I think it is suitable for publication with some minor alterations and additions.

Pg 2:53-60. Some of the references seem unlinked to the statements made; eg ref 16 is a case report, rather than an epidemiological study that could tell us what proportion of cases are neonatal versus late-onset. Likewise ref 10 is about females not about severe neonatal presentations

Pg 2: Triggering factors. Suggest adding a sentence about unaccustomed protein loading, for example the use of sports supplements

Pg 3. Laboratory workup. Suggest adding a sentence about the possible differential diagnoses when orotic aciduria is detected as OTC is not the only cause.

Pg 3: What is the likelihood of NOT finding a mutation in a biochemically confirmed family? It was around 30% in our lab until recently

Pg 4: Dietetic support. We also find monitoring plasma amino acids to be helpful in the dietetic support side of management.

Pg 5. An additional challenge in transition is the possible use of alcohol and illicit substances. These can increase the risk of decompensation events.

Very few grammatical errors to be fixed.

Author Response

Review 1

Overall this is a well written narrative review of late-onset OTC deficiency. I think it is suitable for publication with some minor alterations and additions.

RESPONSE: thank you for your positive comments 

Pg 2:53-60. Some of the references seem unlinked to the statements made; eg ref 16 is a case report, rather than an epidemiological study that could tell us what proportion of cases are neonatal versus late-onset. Likewise ref 10 is about females not about severe neonatal presentations

RESPONSE: The reference no 16 and no 10 were replaced with

Brassier A, Gobin S, Arnoux JB, Valayannopoulos V, Habarou F, Kossorotoff M, Servais A, Barbier V, Dubois S, Touati G, Barouki R, Lesage F, Dupic L, Bonnefont JP, Ottolenghi C, De Lonlay P. Long-term outcomes in Ornithine Transcarbamylase deficiency: a series of 90 patients. Orphanet J Rare Dis. 2015 May 10;10:58. doi: 10.1186/s13023-015-0266-1.

Pg 2: Triggering factors. Suggest adding a sentence about unaccustomed protein loading, for example the use of sports supplements

RESPONSE: added on page 2 in section 2.2

Pg 3. Laboratory workup. Suggest adding a sentence about the possible differential diagnoses when orotic aciduria is detected as OTC is not the only cause.

RESPONSE: A sentence and a reference were added on page 3.

Pg 3: What is the likelihood of NOT finding a mutation in a biochemically confirmed family? It was around 30% in our lab until recently.

RESPONSE: As stated on page 3, the mutation is found in 80-90% [30, 24, 41]. We have added that it may be as high as 30% (personal observation).

Pg 4: Dietetic support. We also find monitoring plasma amino acids to be helpful in the dietetic support side of management.

RESPONSE: We have added it to section 3.2 on page 4

Pg 5. An additional challenge in transition is the possible use of alcohol and illicit substances. These can increase the risk of decompensation events.

RESPONSE: these sentences are now added to the section 3.4. Thank you for the suggestion.

Reviewer 2 Report

Dear author’s

I was pleased to review your manuscript and i have the following comments:

1. The article is a narrative of late-onset OTC deficiency. Please explain the novelty of yoir paper. The articles are published only if they bring new infirmation in the field.

2.  There are no figures or tables  in the manuacript.

3. If is a review type it will be informativ to the readers to introduce a methodology with the review process.

4 Minor punctuation edits.

Author Response

Dear author's

I was pleased to review your manuscript and I have the following comments:

  1. The article is a narrative of late-onset OTC deficiency. Please explain the novelty of your paper. The articles are published only if they bring new confirmation in the field.

RESPONSE: The review is an up-to-date overview of several aspects of late-onset OTC deficiency and the first one written so far. The section about the available and upcoming therapies demonstrates new promising therapeutic options for this rare condition.

  1. There are no figures or tables in the manual.

RESPONSE: this is correct

  1. If is a review type it will be informativ to the readers to introduce a methodology with the review process.

RESPONSE: This is a narrative review discussed by the paediatricians, adult physician, dieticians and a laboratory specialist before we wrote it. The team is involved in the diagnosis and management of the condition as well as clinical trials and therapeutic development for this condition. The literature was reviewed by all the contributing co-authors. It is not a systematic review, so the authors do not think the methodology is necessary. Limited word count is one of the reasons.

  1. Minor punctuation edits.

RESPONSE: These have now been corrected

Reviewer 3 Report

The manuscript cover the most important topics related to the 'Diagnostic and management issues in patients with late onset OTC deficiency'. Nevertheless,thre are several comments for adding few details:

- for medicationt (row 71) that may trigger the hyperammonemia, plese write the examples..steroids and maybe others?

- for the 'Long term management' please include the restricted protein diet (0,8g/Kg -- according to the guides). For long term as well, L-citrulline that is even prefered to Arginine, if available. Beside, regarding the oral nitrogen scavangers..the advantage of glycerol phenylbutyrate - mainly in children

Author Response

The manuscript covers the most important topics related to the 'Diagnostic and management issues in patients with late onset OTC deficiency'. Nevertheless, there are several comments for adding few details:

RESPONSE: Thank you

- for medications (row 71) that may trigger the hyperammonemia, please write the examples..steroids and maybe others?

RESPONSE: These are now listed in section 2.2 and reflect our personal clinical experience: ‘Apart from steroids (dexamethasone, hydrocortisone, methylprednisolone) and chemotherapy, anticonvulsants (valproate and topiramate), metoclopramide and azathioprine may increase demands on the urea cycle, thus triggering an initial OTCD presentation [37].’

- for the 'Long term management' please include the restricted protein diet (0,8g/Kg -- according to the guides). For long term as well, L-citrulline that is even preferred to Arginine, if available. Beside, regarding the oral nitrogen scavengers the advantage of glycerol phenylbutyrate - mainly in children

RESPONSE: Thank you for the valuable suggestions. We have now added them to section 3.2 and section 4.

Reviewer 4 Report

The manuscript entitled “Diagnostic and management issues in patients with late onset OTC deficiency” is interesting and would deserve publication in the journal, children. Before acceptance, however, please revise the comments below.  

1. Line 20: Some cases are due to de novo mutations (for example, see “Ornithine Transcarbamylase Deficiency. https://www.ncbi.nlm.nih.gov/books/NBK154378/ ) and should not declare “X-linked recessive”. I suggest authors to add “in general” or something.  

2. Line 20: “Defects in the OTC gene cause a block in ureagenesis …” is not accurate. “Mutation in the OTC gene cause an impairment in ureagenesis …” may be better.  

3. Line 32: “the body’s primary method” sounds odd. Please revise it such as “the physiological primary pathway”.  

4. Line 38: Please change “Ornithine” to “ornithine”.  

5. Line 75: “occur after a successful diagnosis” sounds odd. Please consider the change to “allowed”, “enabled”, etc.  

6. Lines 168 to 169: The text is duplicative of the above. Please reorganize appropriately.  

7. Lines 187 to 192: Please add the possibility of de novo mutation with proper reference(s).  

8. Line 202: Please explain the abbreviation IMD by “inherited metabolic disorders”.  

9. Lines 241 to 253: The title of the section, mortality does not match the content. Please add a little more about mortality or change the section title.  

10. Lines 276 to 277: The content of the text does not match the content of reference 69. Please change to the appropriate reference.  

11. Lines 346 to 563: Please revise the format of the references to the specified format.

Please improve the text with the cooperation of all the authors.

Author Response

The manuscript entitled “Diagnostic and management issues in patients with late onset OTC deficiency” is interesting and would deserve publication in the journal, children. Before acceptance, however, please revise the comments below. 

  1. Line 20: Some cases are due to de novo mutations (for example, see “Ornithine Transcarbamylase Deficiency. https://www.ncbi.nlm.nih.gov/books/NBK154378/ ) and should not declare “X-linked recessive”. I suggest authors to add “in general” or something.

RESPONSE: it has now been added in the abstract

  1. Line 20: “Defects in the OTC gene cause a block in ureagenesis …” is not accurate. “Mutation in the OTC gene cause an impairment in ureagenesis …” may be better.

RESPONSE: it has now been corrected in the abstract

  1. Line 32: “the body’s primary method” sounds odd. Please revise it such as “the physiological primary pathway”.

RESPONSE: It has now been changed

  1. Line 38: Please change “Ornithine” to “ornithine”.

RESPONSE: It has now been changed

  1. Line 75: “occur after a successful diagnosis” sounds odd. Please consider the change to “allowed”, “enabled”, etc.

RESPONSE: It has now been changed to: ‘…it can only follow a successful diagnosis…’

  1. Lines 168 to 169: The text is duplicative of the above. Please reorganize appropriately.

RESPONSE: it has now been corrected and the repetition has been removed.

  1. Lines 187 to 192: Please add the possibility of de novo mutation with proper reference(s).

It has now been added in section 3.3 and and referenced:

Rüegger CM, Lindner M, Ballhausen D, Baumgartner MR, Beblo S, Das A, Gautschi M, Glahn EM, Grünert SC, Hennermann J, Hochuli M, Huemer M, Karall D, Kölker S, Lachmann RH, Lotz-Havla A, Möslinger D, Nuoffer JM, Plecko B, Rutsch F, Santer R, Spiekerkoetter U, Staufner C, Stricker T, Wijburg FA, Williams M, Burgard P, Häberle J. Cross-sectional observational study of 208 patients with non-classical urea cycle disorders. J Inherit Metab Dis. 2014;37:21–30.

  1. Line 202: Please explain the abbreviation IMD by “inherited metabolic disorders”.

RESPONSE: It has now been changed

  1. Lines 241 to 253: The title of the section, mortality does not match the content. Please add a little more about mortality or change the section title.

RESPONSE: More data on mortality in OTCD and UCDs has been added and referenced accordingly

  1. Lines 276 to 277: The content of the text does not match the content of reference 69. Please change to the appropriate reference.

RESPONSE: The sentence has now been removed and appropriate reference by Morioka et al (2005) has been added on

  1. Lines 346 to 563: Please revise the format of the references to the specified format.

RESPONSE: these have now been updated